# Examining the impact of green marketing practices on business performance: A synergistic application of resource-based view and triple bottom line theory

Md. Asaduzzaman Babu●*, Md. Abdur Rouf, Md. Rohibul Islam, Mahmudul Islam

Department of Marketing, Hajee Mohammad Danesh Science and Technology University, Dinajpur, Bangladesh

* asadbabumkt@hstu.ac.bd

## Abstract

The primary focus of this investigation is to assess the influence of green marketing practices and their potential effects on organizations' operational performance. The factors influencing business performance include environmental, economic, and social elements. Data collection involved 303 respondents who completed a questionnaire utilizing a 5-point Likert scale for each statement. The participants in the study included employees from manufacturing, non-manufacturing, and service sectors. This investigation used a structured equation model (SEM) methodology to assess the impact through the synergic application of RBV and TBL theory, with data analyzed using Smart PLS version 4.1.0.8. The study's findings indicate that implementing green marketing practices has a positive and significant influence on business performance. Additionally, perceived competitive advantage and green innovation marketing strategy are positive mediators in the relationship between green marketing practices and business performance. The findings indicate that business performance substantially impacts environmental, economic, and social performance. The study's uniqueness lies in examining perceived competitive advantage and green innovation marketing strategy as a mediating variable. Businesses are increasingly concentrating on the effects of environmentally friendly marketing strategies on their performance; therefore, this study examines how these practices contribute to enhancing business performance. Future investigations may occur in sectors like FMCG, RMG, leather, and others.

## 1. Introduction

The worldwide marketplace has been drastically changed in the past several years due to the rising demand for sustainable products and the heightened awareness of environmental issues [1,2]. Rising environmental concerns and a sensitive focus

**Data availability statement:** All relevant data are within the paper and its Supporting information files.

**Funding:** The author(s) received no specific funding for this work.

**Competing interests:** The authors have declared that no competing interests exist.

on sustainability have profoundly impacted consumer behavior and global company operations [3]. Companies are under pressure to meet environmental and financial goals. Green marketing has, therefore, become a viable strategy to deal with these issues. Businesses are forced to implement green marketing strategies as consumers are becoming more aware about the environmental influence of the products [4]. The term "green marketing" describes an approach to advertising goods and services that prioritizes minimizing adverse impacts on the environment in line with principles of ethical consumption and sustainable development [5]. By acquiring eco-friendly items, customers may fulfil the objective of mitigating global pollution and safeguarding the environment in alignment with their beliefs [6,7]. Green goods produce less waste and pollution throughout their lives. As individuals grow more eco-conscious, green product demand has surged [8]. Green marketing tactics have emerged as a vital component in contemporary corporate strategy, offering a competitive advantage while tackling environmental issues. Organizations that adopt green initiatives often experience improved brand reputation, stronger customer loyalty, and sustained profitability [9,10,11]. Businesses are prioritizing green products and production methods to combat global warming, implementing clean technology to significantly impact the outcomes of green technological innovation [12]. As worldwide awareness of environmental sustainability increases and customer tastes shift toward eco-friendly solutions, the capacity of companies to develop sustainably is both an operational requirement and a strategic imperative [13]. Concerned individuals worldwide are making climate action a requirement of doing business by demanding that companies get environmental certifications and/or comply with environmental norms. Thus, it is debatable whether consumer demands for more eco-friendly products would prompt businesses to launch green initiatives, which would be good for the environment since they would reduce barriers to entry while simultaneously increasing output [14]. Sustainability-focused businesses have had to adopt new, creative business models to achieve social, environmental, and financial goals while minimizing energy consumption, pollution, and resource depletion [15]. In the traditional context, it may enhance economic performance by lowering expenses and augmenting return on investment via increased profitability, revenue, and productivity [16,17,18]. So, businesses that want to improve their brand's image, boost sales, and save expenses without sacrificing compliance with environmental standards must implement a green marketing plan. In order to remain competitive, appeal to customers concerned about the environment, and help ensure a sustainable future, businesses are embracing green marketing tactics [19,20,21].

A shift from transactional, short-term approaches to relational, long-term ecological frameworks has occurred in marketing theory, driven by the growing focus on sustainability. Brands can benefit from green marketing tactics in several ways, including environmental friendliness, stakeholder trust, and competitive advantage [22,23]. According to the Resource-Based View (RBV), these actions are assets that can strengthen a company's competitiveness and resilience, as they are valuable, rare, hard to imitate, and non-replaceable. Elkington & Rowlands [24] proposed the Triple Bottom Line (TBL), which holds that true success for an

organization extends beyond financial gain and includes social and environmental metrics. By bringing these two perspectives together, we can see how green marketing can boost both non-financial and financial outcomes. So far, the literature has not thoroughly investigated this synthesis. There has been much study of environmental marketing, but some questions remain unanswered. To start, prior studies have primarily examined green marketing from either a consumer or environmental perspective, with an emphasis on buy intent, eco-labeling, or environmental performance outcomes [25,26,27,28]. Research into the organizational mechanisms via which green marketing tactics impact overall company performance is lacking. Secondly, due to an overemphasis on financial and operational efficiency, present research tends to see business success in a simplistic light [29]. This narrow focus contradicts the TBL's all-encompassing framework, which states that sustainable organizational success is defined by economic, environmental, and social performance [6,30,31]. Third, the influence of green marketing approaches on business performance remains unclear due to a lack of precise mediating mechanisms [32]. Previous studies have acknowledged the importance of green practices from a strategic perspective [33,34]. However, no coherent theoretical framework has been developed to empirically assemble the roles of perceived competitive advantages and green innovative marketing tactics as drivers of internal capabilities. Lastly, the majority of the existing research originates from industrialized economies, which have above-average levels of institutional support, technological infrastructure, and consumer environmental consciousness. Inadequate regulatory frameworks, resource constraints, and uneven corporate commitment to sustainability significantly shape the dynamics of green marketing in emerging and developing economies such as Bangladesh. This is why, within these particular constraints, it is crucial to investigate green marketing's function as a resource for improving performance.

This study is significant because of the situation in Bangladesh. The developing nation of Bangladesh is working to green its manufacturing processes. Pollution, resource loss, and industrial waste are becoming increasingly problematic, nevertheless [35]. Textile, leather, and manufacturing companies are under increasing pressure from environmentalists and consumers around the world to improve their practices. In this light, being green with your marketing is not just the right thing to do; it is essential to your business's long-term success and competitiveness on a worldwide scale. However, many Bangladeshi companies lack the know-how, finances, and expertise necessary to implement sustainability-driven marketing strategies. Incorporating sustainability into corporate strategy can be learnt by observing how these corporations utilize green methods to improve their operations. In addition, by showing how sustainability-oriented marketing can deliver business benefits even in resource-limited settings, ideas from this context contribute to the global conversation. This research investigates the association between green marketing practices and business performance, evaluating their impact on financial, environmental, and social results. It also provides helpful insights for organizations seeking to incorporate sustainability into their marketing strategy while achieving long-term growth and profitability. This study's novelty lies in how green marketing practice directly affects business performance, as well as in its mediation by perceived competitive advantage and green innovation marketing strategy.

## 2. Literature review and hypothesis design

Green marketing influences consumer and business behavior by promoting eco-friendly products and activities. Studies have examined its implications on customer behavior, green brand equity, and business model sustainability [36,37]. Eco-innovation creates new ideas, methods, products, and processes to address environmental concerns and attain sustainability. Green innovation marketing that promotes sustainability and innovation helps green businesses prosper [38]. Eco-product innovation increases economic performance and mediates eco-process innovation. Eco-friendly products enhance the economy (Luo et al., 2024). Environmentally sustainable corporate practices reduce business operations' environmental impact and enhance sustainability [39]. Examples include sustainable manufacturing, marketing, HR, investment, and innovation [40]. Green innovations products, services, and processes that promote ecological balance and sustainability are crucial to national development [15].

Environmental labels and green innovation marketing affect customer behavior [41], yet the literature is still lacking. Sustainability, green marketing, and innovative green strategies have primarily been examined in developed nations. Green marketing improves customer behavior and corporate sustainability, raising environmental awareness and firmness [42]. Green marketing is hot, but fresh research suggests its advantages to underdeveloped nations are still being examined. Research in a developing nation shows that green marketing provides pharmaceutical businesses an edge. Green marketing can boost corporate performance in developing countries, but more studies are needed to understand its effects [43].

Current research emphasizes developed nations, making it difficult to comprehend how green marketing, green innovation marketing methods, and perceived competitive advantage interact in developing economies with different limits and potential. A specialized study is needed to investigate how green innovation marketing tactics and perceived competitive advantage might improve company performance and encourage sustainable practices in Bangladesh's industrial environment and consumer behavior. These traits assist the Bangladeshi government and business leaders develop green marketing strategies for sustainable growth.

### 2.1. Role of green marketing

Many businesses use environmentally conscious marketing techniques to improve their overall company success [15]. According to Braik et al. [44], green marketing refers to an organization's commitment to producing safe, eco-friendly goods and services through the use of recyclable and easily decomposable packaging, enhanced pollution control methods, and more cost-effective energy use. Green marketing strategy advocates for ecologically sustainable goods and activities, essential in influencing customer attitudes and guiding company objectives. Researchers (Ismail, 2022; Nguyen-Viet, 2022) examined its influence on customer behavior, characteristics of green brand equity, and the comprehensive incorporation of sustainability into business models. Green marketing practices may satisfy the increasing consumer need for environmental preservation and sustainability by aligning with customers' expectations and beliefs, and companies can enhance consumer loyalty [45,46]. Customer satisfaction thus enhances their market share and income. Personnel exhibiting responsible conduct voluntarily assume extra duties to augment company achievement and boost societal welfare [47]. Green marketing practices significantly influence company performance directly and indirectly [43]. Adopting a green marketing strategy improves business performance and has a good impact on corporate image [48]. Increased participation in green marketing by B2B enterprises correlates with enhanced green competitive advantage for these firms in developed European marketplaces [49].

*H₁ Green marketing practice has a positive impact on business performance.*

### 2.2. Perceived competitive advantage as mediator

Competitive advantage is a long-term strategic goal, and exploring it via a strategic green marketing lens represents substantial research need and potential. Competitive advantage is determined by the organization's distinct position relative to its rivals, achieved using resources [50]. The tactics and qualities that allow a company to achieve consistently better performance in the market are what are known as a sustainable competitive advantage. It improves organizational effectiveness and leads to beneficial economic consequences; it is also essential for valuing products and services [51]. According to Papadas et al. [22], internal green marketing initiatives have a moderating role in the creation of a sustainable competitive advantage. Specifically, the findings corroborate what is already known in green marketing literature: that a strong relationship between strategy and people is a key factor in gaining an edge over the competition. Proactive green marking programs boost competitive advantage, according to research by Yang et al. [52], which found that these programs led to cost savings, improved quality, and more process and product innovation. According to Sheykhan et al. [51], a company's long-term performance is greatly influenced by elements such as policies and resources, quality and delays, motivation and standards, management and surrounds, and government and business. It stresses the requirement of safe procedures

and good environmental management to be ahead of the competition. According to Manigandan and Raghuram [53], the link between a green entrepreneurial strategy, a green company culture, and a competitive advantage is boosted by green innovation. In order to get a competitive edge in the hotel industry, the research suggested that green innovation, an entrepreneurial mindset, and a strong organizational culture are all important factors. Green marketing practices significantly influence company performance directly and indirectly, with sustainable competitive advantage fully mediating the relationship between green marketing approach and business success [54,43,55]. Perceived competitive advantage demonstrates how companies utilize distinctiveness, customer attractiveness, and strategic excellence to demonstrate the value they place on their green marketing initiatives. Based on the above literature, the perceived competitive advantage derived from sustainable practices is a vital mechanism by which environmental activities improve organizational outcomes. Green practices may not significantly enhance corporate performance unless they confer competitive benefits.

*$H_2$ Perceived competitive advantage has positively mediated the relationship between green marketing practices and business performance.*

### 2.3. Green innovative marketing strategy as mediator

In today's highly competitive market, sustainability has moved from a social issue to an important part of business strategy [56]. The most important aspect of this change is the introduction of green innovation, which includes environmentally friendly activities and technology ([57,58]). Green innovation and marketing tactics denote ecologically friendly processes and promotional approaches to advocate for sustainable goods [59,60]. He et al. [38] found that green innovation, which promotes innovation and sustainable practices, greatly increases the success of environmentally friendly businesses. It demonstrates the company's ability to generate innovative ideas for eco-friendly product design, value propositions, communication, and delivery. RBV sees innovation as a critical corporate competency that connects strategy to performance. Green marketing methods often encourage innovation [61]. So, green innovative marketing strategy is the process of transforming green inputs into new outputs that enhance business operations. Environmentally friendly product innovations have a positive effect on economic performance and mediate the effect of environmentally friendly process innovations on economic performance. Luo et al. (2025) found that when companies innovate environmentally friendly products, it boosts the economy. Green innovation and marketing tactics can affect environmental labelling programs and procedures to encourage sustainable purchase behaviors, according to Huang et al. [40]. Innovations in green marketing have a favorable effect on organizational and process innovations, which in turn improve social and environmental performance. Crucially, the study found a strong positive association between financial success and environmental sustainability, indicating that being responsible for the environment may also bring financial rewards [56]. Green practices, particularly in manufacturing, marketing, and investment, have a favorable and substantial effect on firms' economic success and green innovation. Green innovation makes connecting green investment, green marketing, and economic success easier [62]. Khan et al. [63] demonstrated that technology orientation and innovation capability play a significant role in improving green innovation performance, which fosters competitive advantage and enhances economic performance.

*$H_3$ Green innovative marketing strategy has positively mediated the relationship between green marketing practice and business performance.*

### 2.4. Influence of business performance on environmental, economic and social performance

Environmental factors are crucial to sustainability; thus, businesses invest in environmental performance to meet SDGs on environmental quality [64]. It provides crucial information for legislators and international agencies trying to solve environmental sustainability issues in a growing global setting [65]. Reducing consumer-side environmental consequences requires an increasingly important role for ecologically friendly practices [66]. The environmental performance will influence the impact of capital on a firm's sustainability. The financial implications for sustainability-oriented businesses are

compelling. Estimates of financial returns by investors influence the sustainability prospects of organizations [67]. Using data from the tourist industry, Tan et al. [68] looked at how environmental performance, both in aggregate and at the individual level, affected financial performance. According to the results, environmental performance has a positive effect on hotel financial performance. The mediating role of green organizational culture in the connection among CSR, transformational leadership, and ecological results was studied by Khaddage-Soboh et al. [69]. Their research offers useful information for making manufacturing more environmentally friendly. A company's performance may be enhanced by incorporating environmental accounting practices, which help to boost green supply chain management procedures and skillfully handle import restrictions [70]. Positive effects of CSR and environmentally conscious actions on the bottom line of sustainable companies were found by Danish et al. [71]. Another thing they found is that the correlation between CSR and sustainable business success is moderated by environmental performance. Additionally, Mustapha et al. [72] found that economies benefit from prosperous enterprises.

$H_4$ Business performance has a positive influence on environmental performance.

A company's economic performance indicates how effectively it operates and accomplishes its objectives [73]. It is possible to measure it over a short or extended period of time, and it may contain economic development, security, and efficiency indices. Educating company owners implementing practices and principles related to eco-friendly financing, can help build resilience in promoting ecological awareness (Rehan et al., 2024 [74]). According to Rodríguez-Espíndola et al. [75], the main takeaway is that a circular economy, which encourages innovation focused on sustainability, has a favorable effect on financial, environmental, and social outcomes. Wagner [76] found that sustainable management and financial results go hand in hand. In addition, they demonstrated that the performance of the social and environmental systems influenced the economy's performance. According to Mandas et al. [77], the integration of sustainability standards and economic growth significantly affects firms' performance and financial performance. Naseer et al. [78] also revealed a positive correlation between the success of enterprises and the economy's overall performance.

$H_5$ Business performance has a positive influence on economic performance.

As businesses work toward achieving sustainable performance that is more impactful and socially relevant, there is a growing need for enterprises to adopt socially responsible policies and practices [79]. Advocating for environmentally sustainable practices yields benefits for companies, government, and society. A company's bottom line is affected by how well it does in terms of sustainability, which includes financial, ecological, and social metrics [80]. The government allocates fewer resources to pollution control of natural assets like water and air, resulting in societal advantages from a healthy environment and sustainable goods. Green innovation enhances market value by fostering client pleasure and commitment to eco-friendly goods [81,82].

$H_6$ Business performance has a positive influence on social performance.

## 2.5. Theoretical background and conceptual framework

The firm's resource-based view (RBV) posits that a company can outperform others by leveraging its technical, human, and other resources [83]. It is crucial to optimize the capabilities and knowledge of individuals when they are regarded as a critical resource and to prevent their exodus through the implementation of ergonomics [84,85]. The RBV theory endeavors to achieve a sustained competitive advantage by selecting and enhancing resources that are valuable, scarce, costly to replicate, and exploitable by the organization [86,87,88]. Ergonomics can enhance the utilization of valuable, scarce, and expensive human resources, thereby achieving a sustained competitive advantage and business performance that exceeds the norm by enhancing ergonomic jobs and workplace design [89]. So, business performance is the ultimate result of green marketing. In the case of green marketing practice, business performance is indicated by

environmental, economic and social performance as per the triple bottom line (TBL) theory, which is a strategic framework used for incorporating sustainable business practices that aid in managing and enhancing sustainable performance [90,91]. Thus, the study incorporated the RBV theory and the TBL theory to conduct the research and develop the conceptual framework mentioned in Fig 1. The elements of RBV and TBL theories are integrated and methodically examined to assess the influence of green marketing practices on business performance using the performance measures.

The framework (Fig 1: Conceptual Framework) proposed that Green Marketing Practices (GMP) serve as a strategic organizational competency that improves firms' Business Performance (BP) both directly and indirectly. According to RBV, companies that take proactive steps to protect the environment, such as promoting green products, designing eco-friendly products, and operating their businesses in an environmentally responsible manner, gain valuable and unique skills. Eventually, GMP is hypothesized to exert a direct positive influence on BP. In addition, the model incorporates two RBV-driven mediating mechanisms, Perceived Competitive Advantages (PCA) and Green Innovative Marketing Strategy (GIMS). Once business performance improves, firms are better able to allocate resources, invest in sustainability initiatives, and deliver outcomes consistent with the Triple Bottom Line. Therefore, instead of conceptualizing TBL dimensions as internal components of BP, the framework treats them as distinct organizational outcomes that occur after improvements in BP.

## 3. Research methodology

### 3.1. Instrument development

The questionnaire consists of two sections: one describing the respondents' demographic profile and the other asking questions for empirical analysis. Seven constructs, each with four items, were used to conduct the study. These constructs were sourced from previous studies, as mentioned in Table 1 and the items mentioned in S1 Appendix.

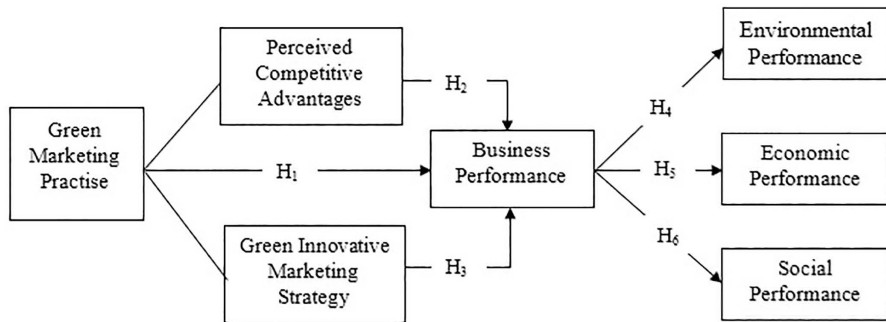

**Fig 1. Conceptual framework.**

**Table 1. Constructs and sources.**

| Variables | Item Code | Number of Item | Source |
|---|---|---|---|
| Green Marketing Practice | GMP | 4 | Chen et al. [15] |
| Perceived Competitive Advantages | PCA | 4 | Chen et al. [15] |
| Green Innovative Marketing Strategy | GIMS | 4 | Tan et al. [92] |
| Business Performance | BP | 4 | Chen et al. [15] |
| Environmental Performance | ENP | 4 | Chen et al. [15] |
| Economic Performance | ECN | 4 | Chen et al. [15] |
| Social Performance | SOP | 4 | Chen et al. [15]; Hermundsdottir & Aspelund [93] |

## 3.2. Sampling procedure

To ensure that the study only included companies that were genuinely implementing Green Marketing Practices (GMP), we employed a multi-step verification method when selecting the sample. If a company met at least two of the following criteria, it was called a "green enterprise": (1) it had environmental certification; (2) it published reports on sustainability or environmental responsibility; (3) it had proof that it had adopted green initiatives; or (4) it took part in recognized environmental programs. We verified these criteria by reviewing the companies' official websites, publicly available reports, and follow-up questions asked at the beginning of the poll. The final sample only included companies that met the eligibility requirements. This ensured that the study only examined companies that were actually using green marketing methods and could provide proof. Researchers use sampling to choose a representative selection of goods or people from a population. The selected people are then monitored or tested to achieve the study's goals [94]. For the developmental research non-probability convenience sampling are used, because they are practical and expected to last. A high probability sample costs and insufficiency for developmental research add to this issue. Therefore, researchers should use convenience sampling [95]. Nonprobability convenience sampling is used in this quantitative investigation. The questionnaire is organized with no open-ended questions. The outcome was based on a five-point Likert scale from strongly disagree (1) to agree strongly (5).

Hair et al. [96] recommends a sample size of five times the number of questionnaire items for this type of investigation. PLS works well with tiny samples but should not priorities be meeting the minimal sample size requirement. Previous research suggests starting route modelling with 100–200 samples [97]. This study comprises 303 respondents based on the last support literature to assure reliability and validity.

## 3.3. Data collection

We disseminated 400 questionnaires across various enterprises (manufacturing, non-manufacturing, and service sectors). 356 completed surveys were received, of which 303 responses were deemed suitable for data analysis. Consequently, 53 responses were eliminated due to absent values and inadequate information. The questionnaires were distributed in mid-November 2024, and the data collection process was completed in January 2025. It took around three months to get all the data. This study was conducted in full compliance with the ethical standards of research. All procedures involving human participants were conducted according to the ethical principles outlined in the Declaration of Helsinki [98]. The data collection revealed a certain message regarding the goal and application of the data. The questionnaire makes clear that each respondent's answers will be used for research goals, guaranteeing no invasion of privacy. The respondents agreed and expressed their view based on the remark. All participants thus gave informed permission for inclusion before starting the research.

Table 2 represents a substantial majority of male respondents, 82.51% of the total, whilst female respondents constitute just 17.49%. No responses are below the age of 20. A significant percentage of respondents, 45.54%, belong to the 31–40 age group. The age distribution indicates that the majority of respondents are middle-aged or elderly. A significant 79.54% possess a post-graduation qualification, establishing it as the most prevalent educational attainment, whereas most respondents originate from a business background (54.79%). The lowest income bracket (below 20,000 Taka) constitutes just 3.96% of the respondents. Individuals earning between 21,000 and 40,000 Taka constitute 31.35%. The predominant group earns between 41,000 and 60,000 Taka, accounting for 35.97% of the sample. A notable percentage (28.71%) earns above 60,000 Taka. This indicates a relatively even distribution among mid- to high-income brackets. Most respondents, including 43.56% of the total, work in Manufacturing organizations. Employment in the service industry is significant, comprising 36.96%. 19.47% are from non-manufacturing entities, which indicates a robust presence in the manufacturing and service industries. The sample included enterprises that have adopted green marketing or sustainability-oriented practices. Questionnaires were administered to middle- and senior-level managers who possess adequate knowledge of their firm's environmental, social, and economic performance.

**Table 2. Demographic profile of the respondents.**

| Categories | | Frequency | Percentage | Cumulative % |
|---|---|---|---|---|
| **Gender** | Male | 250 | 82.51 | 82.51 |
| | Female | 53 | 17.49 | 100.00 |
| **Age (Years)** | Below 20 | 0 | 0.00 | 0.00 |
| | 21-30 | 16 | 5.28 | 5.28 |
| | 31-40 | 138 | 45.54 | 50.83 |
| | Above 40 | 149 | 49.17 | 100.00 |
| **Education** | Graduate | 62 | 20.46 | 20.46 |
| | Post Graduate | 241 | 79.54 | 100.00 |
| | MPhil | 0 | 0.00 | 100.00 |
| | PhD | 0 | 0.00 | 100.00 |
| **Education Background** | Science | 118 | 38.94 | 38.94 |
| | Business | 166 | 54.79 | 93.73 |
| | Social Science | 16 | 5.28 | 99.01 |
| | Others | 3 | 0.99 | 100.00 |
| **Income (Taka)** | Below 20 | 12 | 3.96 | 3.96 |
| | 21000-40000 | 95 | 31.35 | 35.31 |
| | 41000-60000 | 109 | 35.97 | 71.28 |
| | Above 60000 | 87 | 28.71 | 100.00 |
| **Organization Type** | Manufacturing | 132 | 43.56 | 43.56 |
| | Non-Manufacturing | 59 | 19.47 | 63.04 |
| | Service | 112 | 36.96 | 100.00 |

Source: Survey Report 2024–2025.

### 3.4. Participant's consent

Before data collection, written informed consent was obtained from all participants. Each participant was provided with a consent form detailing the study objectives, confidentiality, and voluntary participation. There was a question immediately after the statement about whether you agree to share your opinion; if you did, mark yes; if not, you do not need to fill out the questionnaire. The respondents agreed and expressed their views in response to the remark. Questionnaires were collected and securely stored in a password-protected Google Drive folder accessible only to the researchers: both physical copies (for offline responses) and scanned versions. Only respondents who provided written consent were included in the study.

### 3.5. Measurement model

Formative assessment views measures as causes rather than effects of a construct [99], in contrast to reflective assessment that typically demonstrates the relationship between concepts and evaluation items [100]. To test the validity of the constructs, it is necessary to comprehend their internal consistency, which can be assessed using measures like composite reliability and Cronbach's alpha. Analysis of loadings and average variance recovered is also crucial for evaluating convergent validity. Evaluating discriminant validity is also very important (Hair et al., 2016). Essential component identification, complicated hypothetical interaction analysis, and factor correlation strength evaluation are all tasks that lend themselves well to structural equation modelling (SEM) approaches. Researchers can evaluate the total impact of predictor factors on the outcome variable with these techniques, which include employing a structured model that includes several items and constructs [101,102]. For causal-predictive analysis in extremely complicated scenarios with little theoretical knowledge, Vinzi et al. [103] proposed Partial Least Squares Structural Equation Modelling (PLS-SEM)

as a possible excellent method to do the analysis. While SPLS can be useful in situations when formal theory and a large enough sample are not available, Amos fails to provide an accurate model fit. They concluded that "Both methods are complementary, not competitive." [104]. The purpose of the study dictates the methodology. In order to corroborate or refute the current idea, CB-SEM is selected.

None the less, PLS-SEM excels in both theory building and prediction. Several other research, however, have opted to employ PLS-SEM rather than the covariance technique due to its apparent superiority [105,106]. As a result, Smart PLS version 4.1.0.8 is utilized to assess the cause-and-effect connection in the study, which is based on a reflective-formative measurement model and a SEM technique.

## 4. Empirical analysis

Table 3 displays the results of a Common Method Bias (CMB) test, which examines data to ascertain possible bias in the evaluated constructs. The threshold for this test is 3.3; scores beyond 3.3 imply bias, while values at or below 3.3 signify appropriate levels of common method bias [107]. All reported values are well below the threshold of 3.3, suggesting the absence of substantial common method bias in the evaluated constructs. The results provide significant measurement with no common method bias.

Convergent validity and internal consistency are examined in Table 4. Composite Reliability (CR) and Cronbach's Alpha (CA) assess measurement model internal consistency. CA measures construct dependability, while CR measures construct representation by indicators [108]. The measurement model's internal consistency is sufficient if each construct's CR and CA exceed 0.70 [109,110]. Additionally, Table 4 illustrates indication loadings between 0.765 and 0.976. Data with a cut-off value of 0.708 indicates construct elements fulfil convergent validity standards [105]. Convergent validity is the set of indications used to measure a concept. The average variance extracted (AVE) tests convergent validity by measuring correlation across items intended to reflect the same underlying concept. Convergent validity is established when constructs have an AVE value of 0.50 or above [111]. AVEs for each component in this research vary from 0.718 to 0.922.

Table 5 shows the Fornell-Larcker Criterion, which compares the square root of Average Variance Extracted (AVE) values to the correlation to assess structural equation modelling construct discriminant validity. The variance explained by a construct is shown by these values. Higher values indicate discriminant validity. Thus, discriminant validity is demonstrated when a construct's square root of AVE (diagonal value) is greater than its correlation (off-diagonal values) with any other construct [112]. This table's value shows that each construct meets discriminant validity because their square roots of AVE values are more significant than inter-construct correlations, indicating that all constructs are different and capture different model aspects.

Discriminant Validity Table 6 employs the Heterotrait-Monotrait Ratio (HTMT) to evaluate the distinctiveness of components inside a model. Values under 0.90 indicate strong discriminant validity among constructs [111]. Most HTMT values

**Table 3. Common Method Biased (CMB) Test: Collinearity Statistics (VIF).**

| Constructs | BP | ECP | ENP | GIMS | GMP | PCA | SOP |
|---|---|---|---|---|---|---|---|
| BP | | 1.000 | 1.000 | | | | 1.000 |
| ECP | | | | | | | |
| ENP | | | | | | | |
| GIMS | 1.382 | | | | | | |
| GMP | 1.163 | | | 1.000 | | 1.000 | |
| PCA | 1.379 | | | | | | |
| SOP | | | | | | | |

Source: Author's research results/contribution.

**Table 4. Construct's reliability and validity test.**

| Constructs | Item Code | Convergent Validity | | Internal Consistency | |
|---|---|---|---|---|---|
| | | Loading > 0.70 | AVE > 0.50 | Cronbach's alpha > 0.70 | CR > 0.70 |
| Business Performance (BP) | BP1 | 0.765 | 0.736 | 0.880 | 0.885 |
| | BP2 | 0.926 | | | |
| | BP3 | 0.869 | | | |
| | BP4 | 0.865 | | | |
| Economic Performance (ECP) | ECP1 | 0.963 | 0.92 | 0.971 | 0.972 |
| | ECP2 | 0.951 | | | |
| | ECP3 | 0.973 | | | |
| | ECP4 | 0.949 | | | |
| Environmental Performance (ENP) | ENP1 | 0.883 | 0.805 | 0.919 | 0.924 |
| | ENP2 | 0.888 | | | |
| | ENP3 | 0.924 | | | |
| | ENP4 | 0.893 | | | |
| Green Innovative Marketing Strategy (GIMS) | GIMS1 | 0.940 | 0.821 | 0.926 | 0.956 |
| | GIMS2 | 0.963 | | | |
| | GIMS3 | 0.768 | | | |
| | GIMS4 | 0.941 | | | |
| Green Marketing Practice (GMP) | GMP1 | 0.966 | 0.922 | 0.972 | 0.974 |
| | GMP2 | 0.965 | | | |
| | GMP3 | 0.976 | | | |
| | GMP4 | 0.933 | | | |
| Perceived Competitive Advantages (PCA) | PCA1 | 0.866 | 0.718 | 0.871 | 0.892 |
| | PCA2 | 0.887 | | | |
| | PCA3 | 0.807 | | | |
| | PCA4 | 0.827 | | | |
| Social Performance (SOP) | SOP1 | 0.781 | 0.858 | 0.944 | 0.978 |
| | SOP2 | 0.970 | | | |
| | SOP3 | 0.968 | | | |
| | SOP4 | 0.973 | | | |

Source: Author's research results/contribution.

**Table 5. Discriminant validity: Fornell-Larcker Criterion.**

| Constructs | BP | ECP | ENP | GIMS | GMP | PCA | SOP |
|---|---|---|---|---|---|---|---|
| BP | **0.858** | | | | | | |
| ECP | 0.335 | **0.959** | | | | | |
| ENP | 0.653 | 0.172 | **0.897** | | | | |
| GIMS | 0.525 | 0.128 | 0.661 | **0.906** | | | |
| GMP | 0.479 | 0.272 | 0.605 | 0.326 | **0.960** | | |
| PCA | 0.393 | 0.333 | 0.510 | 0.496 | 0.323 | **0.847** | |
| SOP | 0.595 | 0.164 | 0.462 | 0.198 | 0.432 | 0.081 | **0.926** |

**Note:** The **bold** diagonal numerals represent the square roots of AVE.

**Table 6. Discriminant validity: Heterotrait-Monotrait Ratio (HTMT) Matrix.**

| Constructs | BP | ECP | ENP | GIMS | GMP | PCA | SOP |
|---|---|---|---|---|---|---|---|
| BP | | | | | | | |
| ECP | 0.360 | | | | | | |
| ENP | 0.708 | 0.184 | | | | | |
| GIMS | 0.569 | 0.140 | 0.714 | | | | |
| GMP | 0.500 | 0.279 | 0.643 | 0.339 | | | |
| PCA | 0.412 | 0.361 | 0.547 | 0.549 | 0.350 | | |
| SOP | 0.621 | 0.185 | 0.470 | 0.197 | 0.447 | 0.118 | |

Source: Author's research results/contribution.

fall well below the threshold of 0.118 to 0.714, indicating strong discriminant validity among all components. Discriminant validity was not an issue, as all HTMT values were well below the 0.90 criterion.

The analysis summary (Table 7: Path Analysis) and visualization (Fig 2: Model resolution by PLS algorithm) display the outcomes of a path analysis assessing the interrelations among different constructs. Each pathway is delineated by the Estimates (β), standard deviation (STDEV), T-statistics, and p-values. Estimation (β) = 0.326, which signifies a moderate and favorable direct impact of GMP (Green Marketing Practices) on BP (Business Performance). The T-statistic (8.791) is markedly significant (exceeding the threshold of 1.96), and the p-value demonstrates substantial statistical significance ($P < 0.001$). Estimates (β) = 0.034, which indicates a positive indirect influence of GMP on BP, mediated via PCA (Perceived Competitive Advantage). The T-statistic (2.012) is very significant, above the threshold of 1.96, indicating a statistically significant connection at $P < 0.05$. The mediated impact is less potent than the direct effect ($H_1$), although perceived competitive advantage facilitates the connection between green marketing strategies and enhanced business performance. Estimates (β) = 0.119 indicate GMP's positive and modest indirect influence on BP, mediated via GIMS (Green Innovation Management Strategies). T-statistic = 4.732 and P-value = 0.000, indicating a significant effect ($P < 0.001$). The estimated value (β) of 0.653 signifies BP's robust and favorable influence on ENP (Environmental Performance). The elevated T-statistic (exceeding the threshold of 1.96) and substantial p-value ($P < 0.001$) validate the strength of the relationship. Enhanced business success dramatically contributes to improved environmental performance. The estimated value (β) of 0.335 indicates BP's modest and favorable influence on ECP (Economic Performance). The elevated T-statistic (exceeding the threshold of 1.96) and substantial p-value ($P < 0.001$) validate the strength of the test. Business performance significantly influences economic results, including profitability and cost efficiency. The estimated value (β) of 0.595 indicates BP's robust and beneficial influence on SOP (Social Performance). The exceptionally elevated T-statistic ($10.342 > 1.96$) and the substantial p-value ($P < 0.001$) support the robustness of the association of improved corporate performance results in superior societal outcomes.

**Table 7. Path analysis (Summary).**

| Hypothesized Path | Estimates (β) | Standard deviation (STDEV) | T statistics (|O/STDEV|) | P values | Result |
|---|---|---|---|---|---|
| $H_1$: GMP -> BP | 0.326 | 0.037 | 8.791 | 0.000*** | *Significant* |
| $H_2$: GMP -> PCA -> BP | 0.034 | 0.017 | 2.012 | 0.044** | *Significant* |
| $H_3$: GMP -> GIMS -> BP | 0.119 | 0.025 | 4.732 | 0.000*** | *Significant* |
| $H_4$: BP -> ENP | 0.653 | 0.037 | 17.56 | 0.000*** | *Significant* |
| $H_5$: BP -> ECP | 0.335 | 0.052 | 6.425 | 0.000*** | *Significant* |
| $H_6$: BP -> SOP | 0.595 | 0.058 | 10.342 | 0.000*** | *Significant* |

Note: *** = $P < 0.001$, ** = $P < 0.05$.

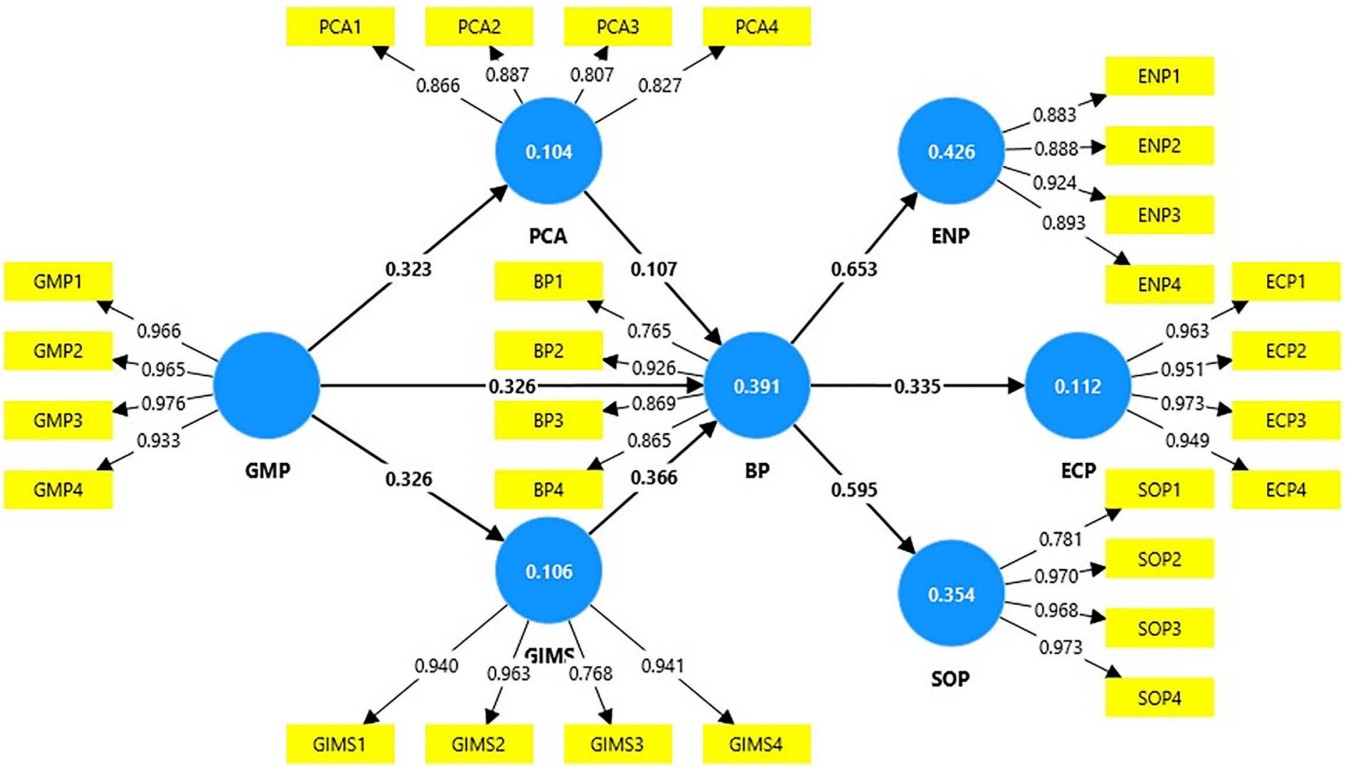

**Fig 2. Model resolution by PLS algorithm.**

Consequently, the investigation validated the acceptance of hypotheses $H_1$, $H_2$, $H_3$, $H_4$, and $H_5$. Green Marketing Practices (GMP) directly and substantially enhance Business Performance (BP) ($H_1$). Business Performance (BP) significantly impacts Environmental Performance (ENP), Economic Performance (ECP), and Social Performance (SOP) ($H_4$, $H_5$, $H_6$). The influence of GMP on BP is facilitated by Perceived Competitive Advantage (PCA) ($H_2$) and Green Innovation Management Strategies (GIMS) ($H_3$). These mediators improve the overall association between GMP and BP, while the direct effect is the most significant. The route analysis demonstrates substantial and strong correlations among green marketing approaches, business performance, and sustainability results. Both direct and mediated consequences emphasize the necessity of implementing sustainable practices and innovative tactics to attain holistic performance enhancements in environmental, economic, and social performance.

## 5. Discussion

This study provides valuable insights into how green marketing practices (GMP) relate to business performance (BP), along with the subsequent effects of BP on environmental, economic, and social performance. There is a direct relation between GMP and BP, proving that sustainable marketing practice positively impacts business performance. This finding aligns with prior research emphasizing the strategic role of environmental sustainability in increasing market competition and customer loyalty [113,114,115]. Organizations can improve financial and operational performance by adopting sustainable marketing strategies to meet the rising customer requirement for responsible environmental behavior. In the Mediated relationships, Perceived Competitive Advantage (PCA) and Green Innovation Management Strategies (GIMS) show that GMP's effect on BP is, thereby, increased through customers and innovation. These observations confirm the assertions highlighting an interaction among sustainability initiatives, consumer engagement and innovations [116]. PCA

requires adapting to shifting customer preferences and involving them in creating value, while GIMS emphasizes constant innovation to achieve sustainability goals. The significant direct effects of BP on ENP, ECP and SOP indicate that business performance plays a vital role in furthering sustainability. These align with resource-based view (RBV) and triple bottom line (TBL) theory, which states that successful firms (financially) can use the availability of resources to address environmental and social challenges ([117,118]). The significant relationship between BP and ENP indicates that economic growth can contribute to environmental actions such as reducing emissions and wastes [119]. Likewise, the impact on ECP and SOP are positive.

This study demonstrates that adopting environmentally friendly production methods transcends mere regulatory compliance; it enhances efficiency, fosters innovation, and results in sustained profitability. The roles of perceived competitive advantage and green marketing strategies also show that a company's skills are important for success [120,121]. Companies that use both green manufacturing methods and strong processes, along with new systems, are more likely to achieve great results. These results are helpful for managers because they show that when environmental efforts match a company's strengths, they can be powerful tools for getting ahead of the competition instead of just being seen as costs [122]. The benefits also apply to society as a whole. Best practices have a significant effect on environmental performance, demonstrating that successful businesses can invest in reducing pollution, conserving energy, and minimizing waste, all of which directly benefit the environment [123,124]. The good link to economic performance also has other benefits, such as creating jobs, steady market growth, and responsible economic contributions. Finally, the strong link between best practices and social performance shows that companies that do better are better able to invest in their employees' health, community development, and corporate social responsibility efforts. This strengthens the notion that attaining organizational success and fostering societal advancement can coexist as complementary objectives rather than conflicting aims. The study effectively demonstrates how GMP serves as a crucial link between business success and the overall health of society. Although some path coefficients, such as GMP ◊ BP ($\beta=0.326$), may appear modest, their practical significance remains noteworthy. Green marketing practices typically require long-term investment, capability development, and organizational cultural change; therefore, a moderate effect size indicates that even incremental improvements in GMP can meaningfully enhance firms' competitive and operational outcomes. Moreover, the relatively strong path from BP ◊ Environmental Performance ($\beta=0.653$) demonstrates that gains in business performance translate into substantial improvements in sustainability outcomes. This suggests that when firms achieve stronger internal performance, they are better positioned to allocate resources to eco-friendly initiatives, adopt clean technologies, and implement environmental management systems. Thus, despite moderate coefficients, the model reveals a cumulative and strategically important impact, particularly in emerging-market contexts where firms often face financial and structural constraints that make even moderate improvements highly valuable.

## 5.1. Theoretical contribution

This research uses the Triple Bottom Line (TBL) framework to understand better how a corporation performs in terms of sustainability. The idea posits that there are three measures of sustainability performance: economic, social, and environmental. By providing experimental validation of these elements as distinct yet interconnected outcomes of organizational practices, our work contributes to the ongoing theoretical discussion of sustainable business performance. It confirms the multifaceted character of TBL in contemporary enterprises. This research shows how sustainability-oriented outcomes are influenced by a combination of internal competencies (such as eco-innovation and environmental management) and external stakeholder demands, by integrating the Triple Bottom Line (TBL) method with the Stakeholder and Resource-Based View (RBV) theories. This integration provides a more thorough framework for understanding sustainability performance by theoretically connecting firm-level resource management with society value generation. Companies in resource-limited regions can learn a lot about introducing and implementing ecologically friendly practices from developing economies like Bangladesh. This shows that sustainability-driven performance can emerge across many institutional and

cultural settings, thereby improving the theoretical frameworks of TBL and RBV. It is not confined to developed markets either. By clarifying the processes that allow environmentally conscious and socially responsible practices to impact multi-dimensional company performance, these contributions advance theories of sustainability and organizational behavior. Most importantly, this research extends the generalizability of TBL and RBV by showing that sustainability-driven performance can emerge in resource-constrained and emerging economy contexts, such as Bangladesh, and is not confined to developed markets. By explicating the processes through which environmentally conscious and socially responsible practices translate into multi-dimensional organizational performance, this study offers a more nuanced and actionable theoretical lens for sustainability research, advancing both the conceptual and practical frontiers of sustainable business performance.

## 5.2. Practical implications

This study provides significant insights for managers and decision-makers aiming to integrate sustainability with competitiveness. First, the positive impact of green marketing practices (GMP) on business performance (BP) indicates that companies should view green initiatives not only as regulatory compliance measures but also as opportunities for growth. Managers should incorporate eco-friendly practices into the design, marketing, and distribution of their products. This will help the market look more sustainable while also making the business more efficient and profitable. Next, the role of perceived competitive advantages (PCA) as a mediator highlights the importance of adopting green practices to differentiate oneself from competitors. To get ahead of the competition, managers should focus on unique value propositions, such as eco-labeling, green certifications, and sustainable product features. This kind of strategic differentiation not only makes customers trust you more, but it also helps you keep your market share over time. Also, the role of green innovative marketing strategy (GIMS) shows that managers need to include new ideas in their plans for sustainability. Creating new marketing campaigns, leveraging digital platforms to share green stories, and collaborating with stakeholders to develop eco-friendly solutions can all enhance GMP's effectiveness. Innovation keeps environmental programs fresh and able to adapt to new consumer needs and government rules. The strong link between BP and environmental performance (ENP), economic performance (ECN), and social performance (SOP) shows that making money can help people make bigger contributions to sustainability. Managers should reinvest some of the money generated from green projects into cleaner technologies, employee growth programs, and community benefit activities. This plan not only builds trust among stakeholders, but it also creates long-term shared value. Finally, combining GMP, PCA, and GIMS gives managers a straightforward way to align their internal strengths with the outside world's growing calls for sustainability. Companies can improve their performance and help the environment, the economy, and social well-being by seeing green practices as valuable strategic assets.

## 5.3. Policy implications

Governments need to make policies that help businesses adopt environmentally friendly practices. This could involve offering tax breaks for green investments, providing financial incentives to companies that utilize sustainable technologies, or initiating programs to recognize companies leading the way in sustainable manufacturing. These programs not only help businesses save money, but they also help different industries adopt green strategies more quickly. Next, we examine how perceived competitive advantages (PCA) and green innovative marketing strategies (GIMS) serve as intermediaries. This shows that companies really benefit from GMP when they can use it to set themselves apart from the competition and drive innovation. Policymakers can help by creating green certification programs, eco-labeling standards, and innovation grants that enable businesses to demonstrate their efforts to be more environmentally friendly to consumers. This method not only levels the playing field, but it also encourages businesses to keep pushing the limits of their green strategies. The strong connection between BP and environmental (ENP), economic (ECN), and social performance (SOP) also supports the idea that companies that do well financially are better able to help with sustainability goals. So,

 

regulators should make rules that include sustainability in performance evaluations. This way, business success is not just based on profits, but also on how they affect the environment and society. For example, requiring companies to use mandatory sustainability reporting frameworks can make them more likely to share and improve their triple bottom line results. Lastly, the study shows that businesses and the government need to work together. Policymakers ought to create platforms that include businesses, regulators, and civil society so that they can work together to find solutions to environmental problems. This collaborative approach ensures that sustainability policies are not only helpful and widely accepted but also align with what organizations can do and what society needs.

## 6. Conclusion

This study emphasizes the crucial impact of green marketing techniques on improving business performance and their consequent effect on sustainability results. The results underscore the necessity of implementing a comprehensive sustainability strategy that incorporates green marketing, proactive consumer adaption, and innovative management techniques. The robust connections between BP and ENP, ECP, and SOP highlight the potential for financial success in promoting sustainability in environmental, economic, and social spheres. This study enhances green marketing and sustainability literature by illustrating direct and indirect relationships between green marketing strategies and business outcomes. It underscores the relevance of triple bottom line theory and resource-based perspectives in comprehending how corporations might attain sustained success while confronting societal and environmental issues. This research offers a novel perspective on the Resource-Based View (RBV) by conceptualizing green practices as dynamic resources, and simultaneously enhances the Triple Bottom Line (TBL) by demonstrating how performance-oriented strategies can facilitate the attainment of sustainability objectives. It gives managers simple tips on how to use green practices to get ahead of the competition while still meeting their social and environmental responsibilities. From a policy point of view, it shows how important it is to make supportive frameworks, incentives, and collaborative platforms to help different industries adopt sustainable practices more quickly.

One key limitation of this study is that the sample was not equally distributed among sectors. Following a stratified sampling technique and ensuring an equal distribution of samples from each industry would be helpful. Another limitation is that data collection took place from November 2024 to January 2025, considered two years. However, this will not significantly impact generalization, as the study confirms the CMB test. Subsequent studies may investigate these links across many sectors and cultural situations to improve the generalizability of the results. The study did not fully incorporate contextual variables such as firm size, industry sector, and regulatory environment, which may influence both the degree of green marketing implementation and the resulting business performance. Future research is advised to specify clearly which organizational characteristics may act as confounding variables and to include them explicitly as control variables in structural models. Doing so will enhance the precision and robustness of causal inferences. Furthermore, investigating the possible moderating influences of organizational culture or market dynamics may yield profound insights into how companies might enhance their sustainability strategy in fluctuating contexts.

## Supporting information

**S1 Appendix. Constructs and items.**
(DOCX)

## Author contributions

**Conceptualization:** Md. Asaduzzaman Babu, Md. Abdur Rouf, Md. Rohibul Islam, Mahmudul Islam.

**Data curation:** Md. Asaduzzaman Babu, Md. Rohibul Islam, Mahmudul Islam.

**Formal analysis:** Md. Asaduzzaman Babu.

**Investigation:** Md. Asaduzzaman Babu.

**Methodology:** Md. Asaduzzaman Babu.

**Project administration:** Md. Abdur Rouf.

**Resources:** Md. Asaduzzaman Babu.

**Software:** Md. Asaduzzaman Babu.

**Supervision:** Md. Asaduzzaman Babu.

**Validation:** Md. Asaduzzaman Babu.

**Visualization:** Md. Asaduzzaman Babu.

**Writing – original draft:** Md. Abdur Rouf, Md. Rohibul Islam, Mahmudul Islam.

**Writing – review & editing:** Md. Asaduzzaman Babu.

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
