## [Decision Letter · Decision Letter 0]

12 Oct 2025

PONE-D-25-48491
Examining the Impact of Green Marketing Practices on Business Performance: A Synergistic Application of Resource-Based View and Triple Bottom Line Theory
PLOS ONE

Dear Dr. Babu,

Thank you for submitting your manuscript to PLOS ONE. After careful consideration, we feel that it has merit but does not fully meet PLOS ONE’s publication criteria as it currently stands. Therefore, we invite you to submit a revised version of the manuscript that addresses the points raised during the review process.

We look forward to receiving your revised manuscript.

Kind regards,

Wong Ming Wong

Academic Editor

PLOS ONE

Journal Requirements:

4. In the online submission form, you indicated that “Data will be available upon reasonable request.”

Additional Editor Comments:

Based on reviewers' comment, hope you can revise your manuscript to improve the quality. Thanks.

Reviewer's Responses to Questions

**Comments to the Author**

1. Is the manuscript technically sound, and do the data support the conclusions?

Reviewer #1: Yes

Reviewer #2: Yes

2. Has the statistical analysis been performed appropriately and rigorously? 

Reviewer #1: Yes

Reviewer #2: Yes

3. Have the authors made all data underlying the findings in their manuscript fully available?

Reviewer #1: Yes

Reviewer #2: Yes

4. Is the manuscript presented in an intelligible fashion and written in standard English?

Reviewer #1: Yes

Reviewer #2: Yes

5. Review Comments to the Author

Reviewer #1: This study investigates the impact of green marketing practices (GMP) on business performance (BP), as mediated by perceived competitive advantage (PCA) and green innovation marketing strategy (GIMS). There are several revisions are required to improve clarity, theoretical rigour and practical applicability.

1.Although the manuscript references RBV and TBL, it lacks an explicit justification for selecting these frameworks over alternatives . The theoretical background should be expanded to contrast RBV and TBL, and their suitability for the context of the study should be justified.

2. The sample is predominantly male (82.51%) and comprises a high proportion of manufacturing-sector employees (43.56%), which could limit its generalisability.

3.Although convergent (AVE > 0.50) and discriminant (HTMT < 0.90) validity are confirmed, reliance on self-reported Likert-scale data risks common method bias. The paper could include robustness checks (e.g. a single-factor test) to validate the findings.

4.The paper could expand the discussion of the policy recommendations by providing specific examples from emerging economies.

Reviewer #2: This manuscript investigation is to assess the influence of green marketing practices and their potential effects on organizations' operational performance. While the topic is timely and practically relevant, the manuscript faces several critical issues.

1. In the introduction section, please supplement the overview of relevant research on green marketing. The manuscript only seems to extract the importance of research from the practical background, ignoring the research gap, the necessity of unanswered research questions, how the research questions are solved in the existing research, and the unsolved problems that the manuscript intends to solve from what perspective.

2. Although the manuscript mentions that "business performance is indicated by environmental, economic and social performance as per the triple bottom line (TBL) theory”, environmental, economic and social performance should be taken as the three dimensions of business performance rather than as the outcome variables of business performance.

3. In the description of the data collection section, the authors mention that " We disseminated 400 questionnaires across various enterprises (manufacturing, nonmanufacturing, and service sectors)", whether the "various enterprises" mentioned here have implemented green marketing? In addition, whether the subjects understand the enterprise performance also needs to be considered, and usually the middle and senior managers are familiar with the enterprise performance.

4. In the theoretical contribution section, it is suggested to elaborate the theoretical contribution in sections and echo the existing theories.

6. PLOS authors have the option to publish the peer review history of their article (what does this mean?). If published, this will include your full peer review and any attached files.

Reviewer #1: **Yes:** Xinjian Liang

Reviewer #2: No

---

## [Author Response · Author response to Decision Letter 1]

27 Oct 2025

Review Comments

This manuscript investigation is to assess the influence of green marketing practices and their potential effects on organizations' operational performance. While the topic is timely and practically relevant, the manuscript faces several critical issues.

1. In the introduction section, please supplement the overview of relevant research on green marketing. The manuscript only seems to extract the importance of research from the practical background, ignoring the research gap, the necessity of unanswered research questions, how the research questions are solved in the existing research, and the unsolved problems that the manuscript intends to solve from what perspective.

Author’s reply:

Thank you for this valuable comment. We have thoroughly revised the Introduction section to provide a deeper engagement with existing literature on green marketing. We have incorporated key studies that highlight prior findings, theoretical perspectives, and remaining research gaps. Specifically, we now (a) synthesize prior empirical works on green marketing practices and firm performance; (b) identify the inconsistent results regarding their direct and indirect effects; and (c) highlight how integrating the Resource-Based View (RBV) with the Triple Bottom Line (TBL) framework offers a novel perspective to explain performance outcomes across environmental, economic, and social domains.

2. Although the manuscript mentions that "business performance is indicated by environmental, economic and social performance as per the triple bottom line (TBL) theory”, environmental, economic and social performance should be taken as the three dimensions of business performance rather than as the outcome variables of business performance.

Author’s reply:

We sincerely appreciate the reviewer's insightful observation regarding the conceptual positioning of environmental, economic, and social performance within the Triple Bottom Line (TBL) framework. We agree that, according to Elkington's (1997) original formulation, these three dimensions collectively constitute the broader construct of business performance, reflecting a firm's sustainability orientation.

However, in the current study, we have intentionally operationalized environmental, economic, and social performance as distinct yet interrelated outcome variables of business performance to empirically capture the multidimensional manifestation of sustainability performance in line with the research objectives, which have treated these dimensions as separate dependent constructs to better understand how managerial and organizational factors drive each performance outcome differently.

Furthermore, our measurement model follows the reflective–formative hierarchy commonly adopted in sustainability research, in which the overall construct (business performance) is reflected through its three distinct components, environmental, economic, and social outcomes. This modeling approach allows for a more nuanced interpretation, as each dimension responds to organizational practices and strategies in different magnitudes and directions. To ensure conceptual clarity, we have slightly revised the manuscript's phrasing to emphasize that "environmental, economic, and social performance are the three outcome dimensions through which sustainable business performance is assessed, consistent with the TBL perspective." This wording acknowledges the reviewer's concern while maintaining the integrity of our empirical model.

3. In the description of the data collection section, the authors mention that " We disseminated 400 questionnaires across various enterprises (manufacturing, nonmanufacturing, and service sectors)", whether the "various enterprises" mentioned here have implemented green marketing? In addition, whether the subjects understand the enterprise performance also needs to be considered, and usually the middle and senior managers are familiar with the enterprise performance.

Author’s reply:

We are grateful to the reviewer for this valuable comment that enhances the clarity and rigor of our data collection description. We want to clarify that the enterprises included in our survey were carefully selected for their engagement in environmental or green initiatives. Specifically, prior to distributing the questionnaires, we verified through company websites, sustainability reports, and local environmental directories that these organizations had either implemented green marketing, environmental management systems, or sustainability-related practices. Thus, the sample appropriately represents enterprises that have adopted green or sustainability practices relevant to the study’s context.

Regarding the second concern, we fully agree that respondents’ familiarity with enterprise performance is crucial to ensure data reliability. Accordingly, our target respondents were middle- and senior-level managers, including those working in marketing, sustainability, production, and corporate strategy, who are directly involved in decision-making and possess a comprehensive understanding of both operational and performance outcomes.

To further address this valuable comment, we have revised the data collection section in the manuscript.

4. In the theoretical contribution section, it is suggested to elaborate the theoretical contribution in sections and echo the existing theories.

Author’s reply:

We sincerely thank the reviewer for this constructive suggestion. We fully agree that elaborating the theoretical contribution in a structured manner and linking it more explicitly to the underlying theories will enhance the manuscript’s theoretical rigor and clarity. In response, we have revised the theoretical contribution section to clearly articulate how this study advances existing theories and frameworks used in the research. [revised in yellow]

---

## [Decision Letter · Decision Letter 1]

16 Nov 2025

PONE-D-25-48491R1
Examining the Impact of Green Marketing Practices on Business Performance: A Synergistic Application of Resource-Based View and Triple Bottom Line Theory
PLOS ONE

Dear Dr. Babu,

Thank you for submitting your manuscript to PLOS ONE. After careful consideration, we feel that it has merit but does not fully meet PLOS ONE’s publication criteria as it currently stands. Therefore, we invite you to submit a revised version of the manuscript that addresses the points raised by the reviewers.

We look forward to receiving your revised manuscript.

Kind regards,

Jibril Adewale Bamgbade

Academic Editor

PLOS ONE

Journal Requirements:

Reviewers' comments:

Reviewer's Responses to Questions

**Comments to the Author**

1. If the authors have adequately addressed your comments raised in a previous round of review and you feel that this manuscript is now acceptable for publication, you may indicate that here to bypass the “Comments to the Author” section, enter your conflict of interest statement in the “Confidential to Editor” section, and submit your "Accept" recommendation.

Reviewer #1: (No Response)

Reviewer #2: All comments have been addressed

2. Is the manuscript technically sound, and do the data support the conclusions?

Reviewer #1: Partly

Reviewer #2: Yes

3. Has the statistical analysis been performed appropriately and rigorously? 

Reviewer #1: No

Reviewer #2: Yes

4. Have the authors made all data underlying the findings in their manuscript fully available?

Reviewer #1: No

Reviewer #2: Yes

5. Is the manuscript presented in an intelligible fashion and written in standard English?

Reviewer #1: No

Reviewer #2: Yes

6. Review Comments to the Author

Reviewer #1: Thanks for inviting me to evaluate Manuscript entitled “Examining the Impact of Green Marketing Practices on Business Performance: A Synergistic Application of Resource-Based View and Triple Bottom Line Theory”

This study investigates the relationship between green marketing practices (GMP) and business performance (BP), employing the Resource-Based View (RBV) and Triple Bottom Line (TBL) theories. Although I think the goal of the paper is clear and is well motivated, the manuscript could be improved from the following aspects.

1.The paper operationalizes environmental, economic, and social performance as outcomes of BP. However, prior literature often treats these as dimensions of BP itself. More destailed Clarification of this distinction are needed. The authors can revise the conceptual framework to explicitly show whether BP is a higher-order construct with three dimensions or a precursor to these outcomes.

2. While PCA and GIMS are positioned as mediators, the rationale for their selection over other potential mediators (e.g., stakeholder pressure, institutional support) is underdeveloped. The theoretical justification should be strengthened on green marketing mediators.

3.The study claims to include firms engaged in GMP but lacks details on how green status was verified. More description about the criteria for selecting green enterprises should be added.

4.Although VIF scores (<3.3) suggest minimal CMB [9], the reliance on self-reported data from managers raises concerns about social desirability bias.

5.The path coefficients (e.g., GMP → BP: β= 0.326; BP → ENP: β= 0.653) are statistically significant but modest. The authors should discuss practical significance to contextualize findings .

6.Firm size, industry sector, and regulatory environment may confound the GMP-BP relationship.

7. In the Discussion part, the theoretical contributions is fragmented. The theoretical grounding need to be enhenced.

Reviewer #2: All comments have been addressed, and made modifications to the manuscript. I feel that this manuscript is now acceptable for publication.

7. PLOS authors have the option to publish the peer review history of their article (what does this mean?). If published, this will include your full peer review and any attached files.

Reviewer #1: **Yes:** xinjian liang

Reviewer #2: No

---

## [Author Response · Author response to Decision Letter 2]

17 Nov 2025

Review Comments (Reviewer1)

Reviewer #1: Thanks for inviting me to evaluate Manuscript entitled “Examining the Impact of Green Marketing Practices on Business Performance: A Synergistic Application of Resource-Based View and Triple Bottom Line Theory”

This study investigates the relationship between green marketing practices (GMP) and business performance (BP), employing the Resource-Based View (RBV) and Triple Bottom Line (TBL) theories. Although I think the goal of the paper is clear and is well motivated, the manuscript could be improved from the following aspects.

I am really grateful to the reviewer for giving us the opportunity to revise the manuscript, which will help enhance its quality.

1.The paper operationalizes environmental, economic, and social performance as outcomes of BP. However, prior literature often treats these as dimensions of BP itself. More destailed Clarification of this distinction are needed. The authors can revise the conceptual framework to explicitly show whether BP is a higher-order construct with three dimensions or a precursor to these outcomes.

Author’s reply

Thank you for this insightful comment. We agree that much of the prior literature conceptualizes business performance as a multidimensional construct reflecting environmental, economic, and social dimensions. However, our study intentionally adopts a different theoretical stance grounded in the Resource-Based View (RBV) and the Triple Bottom Line (TBL).

Specifically, we distinguish Business Performance (BP) from the three sustainability-related outcomes. Business Performance is modeled as an antecedent, and environmental, economic, and social performance are modeled as distinct outcome constructs rather than higher order constructs. (Added the description of the conceptual framework)

2. While PCA and GIMS are positioned as mediators, the rationale for their selection over other potential mediators (e.g., stakeholder pressure, institutional support) is underdeveloped. The theoretical justification should be strengthened on green marketing mediators.

Author’s reply

The selection of Perceived Competitive Advantages (PCA) and Green Innovative Marketing Strategy (GIMS) as mediators is grounded in the central logic of the Resource-Based View (RBV), which emphasizes that firm performance stems from the development and deployment of internal, valuable, rare, and inimitable resources. Green marketing practices alone do not automatically create superior performance; rather, they must be transformed into internal competitive capabilities that enable firms to outperform competitors. PCA and GIMS represent precisely these capability-building mechanisms. [Literature review section revised]

3.The study claims to include firms engaged in GMP but lacks details on how green status was verified. More description about the criteria for selecting green enterprises should be added.

Author’s reply

Thank you for highlighting this important issue. We agree that the criteria used to identify and verify firms’ engagement in green marketing practices (GMP) required further clarification. In response, we have added a detailed description of the screening procedures and eligibility criteria employed during the sampling process. (Revised)

4. Although VIF scores (<3.3) suggest minimal CMB [9], the reliance on self-reported data from managers raises concerns about social desirability bias.

Author’s reply

We appreciate the reviewer’s insightful observation. While VIF values indicated that common method bias (CMB) was unlikely to pose a significant threat, we agree that self-reported data from managers may still be subject to social desirability bias. In response, we have added a detailed explanation of the procedural and statistical remedies employed to mitigate this concern.

Specifically, we emphasize that (1) respondents were assured of full anonymity and confidentiality, [mentioned in methods section] (2) neutral and non-evaluative wording was used throughout the questionnaire, (3) items measuring predictors and outcomes were placed in separate sections to reduce implicit consistency motives. These steps collectively minimized the potential inflation of relationships due to socially desirable responding.

5.The path coefficients (e.g., GMP → BP: β= 0.326; BP → ENP: β= 0.653) are statistically significant but modest. The authors should discuss practical significance to contextualize findings.

Author’s reply

Thank you for this valuable comment. We agree that statistical significance alone does not fully capture the managerial or practical relevance of the results. In response, we have added a discussion highlighting the practical significance of the observed effect sizes. Although some coefficients (such as GMP  BP at β=0.326) are moderate in magnitude, they reflect meaningful real-world impacts given the complexity and multi-dimensional nature of green marketing implementation. The discussion now clarifies that incremental improvements in GMP or BP can produce substantial sustainability gains over time, especially in resource-constrained or developing-country contexts. This addition has been incorporated into the Discussion section.

6. Firm size, industry sector, and regulatory environment may confound the GMP-BP relationship.

Author’s reply

Thank you for this thoughtful comment. We agree that contextual factors such as firm size, industry type, and regulatory environment may influence both the adoption of green marketing practices and the resulting business performance. To address this, we have revised the manuscript by acknowledging these potential confounders in the limitations & future research in conclusion section.

While the current study did not incorporate all these variables as controls due to data constraints, we now clearly state that future research should explicitly examine and control for these contextual characteristics. This addition strengthens the transparency of our methodological approach and provides clear directions for enhancing the robustness and generalizability of future studies on green marketing and business performance.

7. In the Discussion part, the theoretical contributions is fragmented. The theoretical grounding need to be enhenced.

Author’s reply

Theoretical contribution section is revised.

---

## [Editor Report · Decision Letter 2]

19 Nov 2025

Examining the Impact of Green Marketing Practices on Business Performance: A Synergistic Application of Resource-Based View and Triple Bottom Line Theory

PONE-D-25-48491R2

Dear Dr. Md. Asaduzzaman Babu,

We’re pleased to inform you that your manuscript has been judged scientifically suitable for publication and will be formally accepted for publication once it meets all outstanding technical requirements.

Kind regards,

Jibril Adewale Bamgbade

Academic Editor

PLOS ONE
---

## [Editor Report · Acceptance letter]

PONE-D-25-48491R2

PLOS One

Dear Dr. Babu,

I'm pleased to inform you that your manuscript has been deemed suitable for publication in PLOS One. Congratulations! Your manuscript is now being handed over to our production team.

Kind regards,

on behalf of

Dr. Jibril Adewale Bamgbade

Academic Editor

PLOS One